# Identifying and Predicting Trends of Disruptive Technologies: An Empirical Study Based on Text Mining and Time Series Forecasting

**Minhao Xiang** [1] , **Dian Fu** [2] **and Kun Lv** [1,*]

1. Business School, Ningbo University, Ningbo 315211, China
2. Teaching and Education College, Ningbo University, Ningbo 315211, China
* Correspondence: lvkun@nbu.edu.cn

**Abstract:** Disruptive technologies are related to a country's competitiveness and international status. Accurately identifying and predicting the trends in disruptive technologies through scientific methods can effectively grasp the dynamics of technological development, adjust the national science and technology strategic layout, and better seize the high ground in international competition. Based on patent text data, this paper uses the improved LDA2Vec model combined with relevant indicators to identify the main topics in disruptive technologies, and predicts and analyzes the development trend through the establishment of an ARIMA model. Taking the energy technology field as an example, the main topics and development trends concerning disruptive technologies in this field are obtained. The study found that ten technologies, including energy storage technology, energy internet management technology, and offshore wind energy technology, are disruptive technologies in the energy technology field, and the development speed of energy storage technology is the fastest. To verify the correctness of the conclusion, this paper compares the results with artificial verification methods such as expert interviews and document verification, and finds that the two are basically consistent, thus verifying the effectiveness and feasibility of the proposed method.

**Keywords:** disruptive technology; text mining; topic recognition; trend prediction; energy technology

## 1. Introduction

In recent years, a new round of technological revolution and industrial transformation is restructuring the global technological competitive landscape. This is an opportunity for the development of fossil, new, and renewable energy to take turns replacing each other. Countries around the world are adjusting their technological development strategies, and the phenomenon of "disruption" in the field of energy technology is frequently seen. In this process, disruptive technologies are constantly emerging, having a huge impact and receiving widespread attention, becoming the strategic focus of competition between technological powers. From the perspective of the historical context, numerous significant technological breakthroughs are fundamentally changing social production and the order of human life. The occurrence and breakthrough of technological revolutions are marked by the emergence and maturity of disruptive technologies. Therefore, it is very important to seize the current window period and strengthen strategic research on disruptive technological innovation. In order to do this, China attaches great importance to the development of disruptive technologies and has made a series of responses in national policies and strategic deployment. However, there are currently issues in China's development of disruptive technologies, such as unclear topics and development paths. It is necessary to start with the essence of disruptive technologies, according to their development laws, to use intelligence analysis methods to strengthen the identification of disruptive technology topics, and to deploy scientific and reasonable strategic guidelines according to technological development trends, in order to seize the high ground in international competition. Therefore, faced

with the wide coverage of disciplinary fields regarding disruptive technologies, this paper selects the important field of energy technology for in-depth analysis to achieve accurate identification of disruptive technology topics, helping to grasp the development trend, and providing important support for strategic layout and decision-making in this field.

## 2. Literature Review

The concept of disruptive technology was first proposed by Professor C. Christensen, who believed that certain technologies often enter the market through low-end or niche markets, characterized by simplicity, convenience, and affordability in their initial stages. With the continuous improvement and refinement of performance and functionality, these technologies eventually replace existing technologies, open up new markets, and create new value systems [1]. Disruptive technology typically has a profound impact on traditional industry structures, business models, market patterns, and may even lead to the complete overturning of the existing industrial landscape [2]. Therefore, disruptive technology is referred to as a revolutionary force that "changes the rules of the game" and "reshapes the future pattern", and is a unique innovative behavior that emerges from scientific research and technological development [3]. Research methods regarding disruptive technology include qualitative and quantitative analyses. Qualitative analysis mainly involves in-depth analysis and exploration of the emergence, development, and impact of disruptive technology through literature research, expert interviews, case studies, and other methods. Quantitative analysis mainly involves the quantification and analysis of the trends, impact, and applications of disruptive technology through data analysis, modeling, forecasting, and other methods [4,5]. As the status of disruptive technology continues to rise, research on disruptive technology has become a hot topic in academia. Research on the concept differentiation [6], characteristics [7], identification methods [8], evolution path [9], and innovation mechanisms [10] has made preliminary progress. Among these, research on technology identification and prediction has undertaken a theoretical analysis of disruptive technology and provided strong support for its strategic development, playing a crucial role in the process of technological innovation.

In terms of technological recognition, early recognition of disruptive technologies in specific fields mainly relied on expert experience, but due to the scarcity of expert resources and the narrow scope of application, it was difficult to analyze a large amount of technology and meet the needs of disruptive technology recognition [11]. To solve this problem, scholars have gradually begun to pay attention to text mining methods, and using these methods to study the identification of disruptive technology has become one of the research hotspots [12]. Text mining methods can extract information about disruptive technology from large-scale patent literature, news reports, and social media data. Among these, topic modeling is a commonly used text mining method, which can convert text data into topic distribution representation, thus helping to analyze the trend and evolution of disruptive technology [13,14]. Latent Dirichlet Allocation (LDA) topic models and other natural language processing methods can mine deep-level implicit knowledge of technology, with better recognition effects [15]. For example, Momeni [16] combined patent and literature data, followed the technological development path through patent citation data, and analyzed the core technology theme through topic modeling. By analyzing the changes in the amount of literature, he analyzed the importance and quantity of technological changes. Dotsika and Watkins [17] built a keyword co-occurrence network of academic publications and used node location indicators such as proximity and distance in social network analysis methods to judge keywords that may have disruptive potential. The research found that keywords with high proximity and low degree are most likely to have disruptive potential. Yoon et al. [18] proposed a network-based patent analysis method to study the relationship between patents in a specific field and determine the possibility of technology outbreaks, thus identifying potential disruptive technologies.

In terms of technology forecasting, China's Ministry of Science and Technology included disruptive technology forecasting and evaluation in its *Fifth National Technology*

*Forecast* in 2015. Based on the definition and selection principles of disruptive technology, the forecast results were mainly generated through expert recommendations via interviews [19]. In a general sense, technology identification is a research issue in technology forecasting, which describes the system process of a technology's emergence, performance, characteristics, or impact in the future at a certain time [20]. In this process, data analysis perspectives, integrating subjective and objective data, are necessary to ensure the scientific nature of technology forecasting [21]. Time series analysis is an effective prediction method, and common examples include autoregressive models, moving average models, Autoregressive Integrated Moving Average model (ARIMA), Vector Autoregression model (VAR), etc. These methods can fit the data, find its regularity, and predict future trends, and have gradually been applied to the study of disruptive technology [22]. However, in the process of predicting disruptive technology, it is necessary to deal with multivariate and nonlinear problems, and traditional time series methods may not be competent. To solve these problems, scholars have proposed various time series analysis models and methods based on machine learning and deep learning, including machine learning algorithms, such as support vector machines, random forests, and deep learning algorithms, such as recurrent neural networks and convolutional neural networks [23,24]. These algorithms can automatically learn the characteristics of time series data and adapt to different data types and prediction tasks. In addition, ensemble algorithms such as AdaBoost and Gradient Boosting have also been applied to time series prediction, which can improve prediction performance by combining multiple prediction models [25,26]. For example, Hughes [27] proposed a big data-based technology sequence forecasting model that combines technology sequence analysis and organizational big data tools. The model emphasizes the differentiated roles and timing of front-end translator experts, data scientists, and industry scientists in conjunction with big data in technology forecasting, thus discussing its application in disruptive technology forecasting. Adamuthe et al. [28] used patent and paper time series data and adopted trend forecasting and growth curve methods to select and analyze six computing technologies, determining that mainframes, minicomputers, and cloud computing are disruptive technologies. Based on an evolutionary perspective, Linton [29] introduced the Bass innovation model into the field of disruptive technology forecasting, arguing that it is necessary to separately consider multiple independent markets to obtain more accurate parameter estimates. The prediction results can guide the combined effects of learning curve effects.

In summary, current research on disruptive technology topic recognition and trend prediction has been widely studied by scholars. Research methods have shifted from single expert qualitative judgment to text mining with expert experience. As a result, text mining methods such as LDA models have become increasingly popular in technology topic recognition. However, recent research has found that using LDA models alone has limitations, such as ignoring semantic relationships between words, affecting the accuracy of recognition and hindering subsequent trend prediction. In terms of time series prediction, although some algorithms have shown high predictive performance, they also face problems, such as overfitting, imbalanced data, and missing values, which may lead to a decline in model performance. Therefore, when using these algorithms for prediction, it is important to carefully select appropriate features and models, and conduct proper data preprocessing and model tuning. To address these challenges, this article proposes an improved Latent Dirichlet Allocation to Vector (LDA2Vec) model that combines global semantic word vectors and text-based topic vectors. The LDA2Vec model is a deep learning model that aims to map words to a low-dimensional vector space so that similar words are closer together in the vector space, which can be used to identify the semantic similarity of words and address the limitations of using a single LDA topic model lacking contextual semantic information.. Using this improved LDA2Vec model, the article analyzes patent text data to achieve accurate recognition of disruptive technology topics, and then uses a time series prediction model to complete trend prediction, determining the

future development of technology topics and providing theoretical support to effectively grasp the development of disruptive technologies in related fields.

## 3. Methodology

This article presents a disruptive technology topics identification method that combines the LDA model and Word2Vec model, and predicts the development trend of technology topics through the ARIMA model, including the following five processes. First, determine the type of data to be obtained, collect relevant data through database searches and crawlers, and implement a data format with time tags and text content. Second is text cleaning and preprocessing. Third, construct Word2Vec word vectors and LDA topic vectors, use TF-IDF to calculate word vector weights, construct fusion feature vectors, and combine the K-means algorithm to cluster technology topics. Fourth, identify disruptive technology topics based on disruptive technology feature indicators. Fifth, construct time series for each topic and predict the development trend of disruptive technology topics based on the ARIMA model, as shown in Figure 1.

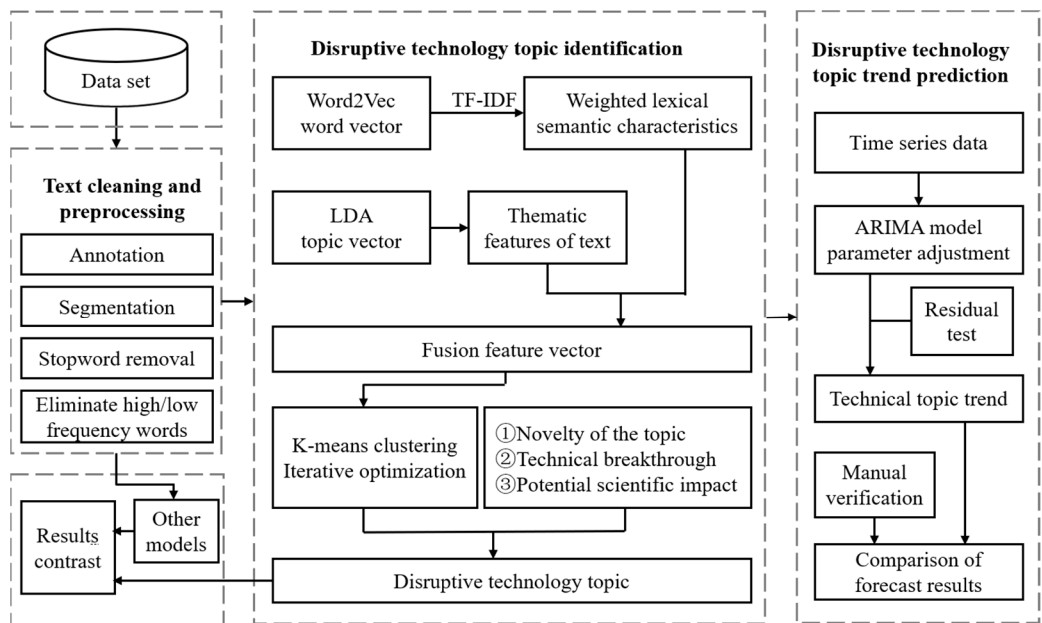

**Figure 1.** Workflow of the proposed research methodology.

### 3.1. Date Set Construction

Text data is the most widely used carrier for studying disruptive technology, and patent abstracts are important references for identifying technology topics. There are generally two methods for obtaining text data: one is to find existing datasets, such as third-party corpus databases, such as Wikipedia; the other is to use crawlers to collect data, e.g., by writing corresponding crawlers based on frameworks such as Beautiful Soup or Scrapy, or by using professional search tools to obtain data. Especially for technology topic identification, in which the research is often focused on a specific field, open corpus databases often cannot meet research needs, so the second method is the main source of text data acquisition.

### 3.2. Text Data Cleaning and Preprocessing

Before actually using text data, it must be cleaned and preprocessed to suit the research needs. Because data collected by crawlers may contain irrelevant information such as HTML tags, it is necessary to remove this to avoid affecting subsequent steps. A small amount of non-text content, special non-English characters, and punctuation marks can be removed using Python's regular expressions.

Tokenization is an important step in text preprocessing, which involves dividing a sentence into multiple independent words. Modern tokenization is based on statistical tokenization, and the statistical sample content comes from some standard corpus. For any two words, $\omega_1$ and $\omega_2$, their conditional probability distributions can be approximated by formulas (1) and (2), respectively:

$$P(\omega_2|\omega_1) = \frac{P(\omega_1, \omega_2)}{P(\omega_1)} \approx \frac{freq(\omega_1, \omega_2)}{freq(\omega_1)} \tag{1}$$

$$P(\omega_1|\omega_2) = \frac{P(\omega_2, \omega_1)}{P(\omega_2)} \approx \frac{freq(\omega_1, \omega_2)}{freq(\omega_2)} \tag{2}$$

where $freq(\omega_1, \omega_2)$ represents the number of times $\omega_1$ and $\omega_2$ appear together in the corpus, and $freq(\omega_1)$ and $freq(\omega_2)$ represent the number of times $\omega_1$ and $\omega_2$ appear in the corpus, respectively. Using the statistical probability established by the corpus, for a new sentence, the optimal word segmentation can be found by calculating the joint distribution probability of various word segmentation methods and finding the one with the highest probability. For the word segmentation function needed in text mining, Python's word segmentation component can be used. For example, English word segmentation can use NLTK.

After word segmentation, stop word processing is needed. Stop words are words that have no actual meaning in a sentence and have no effect on the understanding of the semantics of the whole sentence. In text, there are a large number of meaningless words, such as pronouns, verbs or nouns with no specific meaning. These meaningless words are of no help in text analysis, and removing stop words can reduce the workload of subsequent text processing.

*3.3. Disruptive Technology Topics Identification Based on LDA2Vec*

3.3.1. LDA Topic Vector Construction

The LDA probability topic model has a three-level structure, from top to bottom: document level, topic level, and feature word level. As a three-level Bayesian probability model, LDA can reduce the dimensionality of a segmented document into a topic distribution, and according to the relevant topic probability in the corresponding feature vector, obtain the corresponding document topic, essentially using the co-occurrence feature of text feature words to mine the topic of the text. In doing so, the word vector is trained, and the calculation method is shown in Formula (3):

$$x = \{C(w_1), C(w_2) \ldots C(w_v)\} \tag{3}$$

where $x$ is the word vector representation of all vocabulary, $v$ is the total number of words in the vocabulary dictionary, and $C(w_1)$ is the word vector representation of word $w_1$.

This paper uses the LDA model in the Gensim module to train the text training set for topic modeling. Based on the training results, select the topic $z_{max}$ with the highest probability for document $d_i$, then select the top $n$ words $(t_1, t_2, t_3, \ldots, t_n)$ and their probability values $(p_1, p_2, p_3, \ldots, p_n)$ according to the topic word file, and normalize the probability value as the weight information of the $n$ words. Formula (4) is as follows:

$$q_i = \frac{p_i}{\sum\limits_{a=1}^{n} p_a} \tag{4}$$

where $q_i$ is the normalized value of $p_i$, and $(q_1, q_2, q_3, \ldots, q_n)$ is the weight of the top $n$ words. The weighted sum of the trained word vectors $(C(t_1), C(t_2) \ldots C(t_k))$ is used to obtain the topic vector, and the calculation method is shown in Formula (5):

$$d_z = \sum_{b=1}^{n} q_b \times C(t_b) \tag{5}$$

### 3.3.2. Word2Vec Word Vector Construction

The Word2Vec word vector model is a lightweight neural network model used for text word vector learning, with a three-layer structure of input layer, hidden layer, and output layer. Based on the input and output, the model framework is mainly divided into the CBOW and Skip-gram models. The Word2Vec model mathematically transforms words into dense real-valued vectors in a low-dimensional space, achieving the feature expression of text vocabulary. The word feature vector generated by the model contains semantic associations with adjacent words, which can make up for the semantic deficiency of summary text vocabulary. Therefore, this paper adopts the Skip-gram learning mode to predict the context of the target word. The optimization objective function of the Skip-gram model is shown in Formula (6):

$$L = \frac{1}{T}\sum_{k=1}^{T}\sum_{-c \leq j \leq c} \log p\left(w_{k+j}\middle|w_t\right) \tag{6}$$

where $c$ is the size of the window; the larger the value of $c$, the more training samples obtained, the higher the accuracy of the results, but the longer the training time required.

Given a word $w_k$, the context $w_{k+j}$ of the word $w_k$ is predicted, and the word vector of the patent summary data set is generated. The word vectors are mapped to the vocabulary in the patent summary text, as shown in Formula (7):

$$d_c = \begin{bmatrix} l_{11} & l_{12} & l_{13} & \cdots & l_{1t} \\ l_{21} & l_{22} & l_{23} & \cdots & l_{2t} \\ \vdots & \vdots & \vdots & \cdots & \vdots \\ l_{k1} & l_{k2} & l_{k3} & \cdots & l_{kt} \end{bmatrix} \tag{7}$$

where line $k$ represents the word vector corresponding to the vocabulary $w_k$ in the text, and $t$ represents the dimension of the word vector.

### 3.3.3. Feature Vector Fusion

The paper uses the TF-IDF value to weight the Word2Vec word vectors in order to improve the ability of the word vectors to distinguish topics. This is because the Word2Vec model is unable to represent the degree to which a word contributes to a topic, which can cause non-important words to influence the semantic expression of the features. The higher the TF-IDF value, the more important the word is, and the weight feature is $T = [t_1, t_2, t_3, \cdots, t_k]$, where $t_k$ represents the weight of the word $w_k$ in the text. The weighted vector $d_c' = T \times d_c$ is obtained by multiplying the word vector $d_c$ with its corresponding TF-IDF value.

In this paper, the LDA topic vector $d_z$ and the TF-IDF weighted Word2Vec word vector $d_c'$ are concatenated vertically to form the fused feature vector $doc_m = \{d_z : d_c'\}$, where ":" is the vector concatenation operator. $doc_m$ contains global semantic information, word order information, and deep semantic association information in the text, and compensates for the shortcomings of both LDA topic vectors and Word2Vec word vectors, enriching the semantic information of the summary text vector.

### 3.3.4. Technical Topic Clustering

K-means is a commonly used clustering analysis algorithm that can automatically divide objects into $K$ classes based on the distances between their attributes. By comparing the distances from different $K$ values to their corresponding class centers, the optimal number of classes can be determined. Since the $K$ value of the K-means algorithm needs to be manually set, different K values will produce different results. It has also been observed that the parameter $K$ is equivalent to a parameter that determines the number of topics in a topic model. Therefore, there is a natural connection between modeling topics and

K-means clustering [30]. The optimal number of topics can be determined using perplexity, which in turn can be used to determine the *K* value for the K-means algorithm.

In this paper, the K-means algorithm is used to cluster the fused feature vector $doc_m$ by topic, grouping texts that are similar in content into the same cluster, representing a technical topic. The steps of the K-means algorithm are as follows: select *K* samples as the initial clustering centers $a = a_1, a_2, \ldots, a_k$, for each sample $x_i$ in the dataset, calculate the distance to the *K* clustering centers, and assign it to the class corresponding to the closest clustering center; for each class $a_j$, recalculate its cluster center, which is the centroid of all the samples belonging to that class; repeat the above steps until the set termination condition is met. Formula (8) for calculating the cluster center is as follows:

$$a_j = \frac{1}{|c_i|} \sum_{x \in c_i} x \tag{8}$$

*3.4. Disruptive Technology Topics Measurement Index Determination*

In this paper, the novelty of the topic [31], the technical breakthrough [32], and the potential scientific impact [33] are used as the measurement indicators for disruptive technical topics, and the normalized values of these three indicators are combined to obtain the detection value. The output ranking is based on these values, and the top ranking topics are identified as disruptive technical topics. The specific indicator content is as follows.

3.4.1. The Novelty of the Topic

The emergence of disruptive technology is based on a new technological breakthrough or different combinations of existing technology, making it innovative and reflecting the novelty of the technical topic. Its characteristic is that, when a topic first appears, its novelty is high, but as time goes on and the topic becomes more popular, its novelty gradually decreases. The novelty is calculated by slicing and sorting the topic documents corresponding to the topic according to the time, and taking the first year in which the topic appears as the starting year. The novelty of the topic $z$ in year $t$ is calculated according to Formula (9).

$$NI_t^z = \frac{1}{t - FY + 1} \tag{9}$$

where $t$ represents the current year, $FY$ represents the starting year, and $NI_t^z$ represents the novelty of the topic $z$ in year $t$.

3.4.2. Technical Breakthrough

The technical breakthrough of disruptive technology can measure the progress of the same topic technology. The Organization for Economic Co-operation and Development (OECD) mentions in its reports that the diversity of patent citations can be used to determine the degree of technological breakthrough. Therefore, this paper selects the group of patent technologies with the same International Patent Classification (IPC) classification number and uses the diversity of patent citations to measure the degree of technological breakthrough of disruptive technology, as shown in Formula (10).

$$B = \sum_{i}^{n_p} \frac{CT_i}{n_p} \tag{10}$$

where patent $i$ is the original patent and patent $p$ is the backward reference patent of patent $i$, and their IPC numbers are different. $CT_i$ represents the number of IPC numbers of patent $p$ that reference patent $i$, and $n_p$ represents the number of IPC numbers in all backward references of patent $p$. The higher the $B$ value, the more diverse the technology that the patent depends on and the bigger the breakthrough.

### 3.4.3. Potential Scientific Impact

The potential scientific impact of disruptive technology is reflected in its ability to predict future trends and lead potential. Based on prediction of the future, the ideas conveyed by disruptive technology are more advanced and can guide researchers in adjusting and upgrading their technology via an iterative process to conform to the development trend. In order to avoid deviating from the law of technological development, it is necessary to introduce expert evaluations of technical topics based on objective data. Therefore, the scores of experts for each technical topic need to be counted, and the statistical data for each item is weighted to obtain the overall score of experts for the technical topic. Formula (11) is as follows:

$$R = \frac{\sum\limits_{j=1}^{n} s_{ij} \times r_{ij}}{\sum\limits_{j=1}^{n} s_{ij}} \tag{11}$$

where *i* is the *ith* technical topic, *j* is the *jth* expert, *s* is the evaluation weight given by the expert's familiarity with the field, *r* is the score given by the expert, and *R* is the overall score of different experts for the same technical topic.

### 3.4.4. Disruptive Technology Topics Detection Formula

Through the feature analysis of text data, the three indices of novelty of topic, technological breakthrough, and potential scientific impact are obtained, and then the method of component weight allocation and the idea of multi-index technical topic recognition formula in literature [34] are borrowed to construct a formula for identifying a disruptive technical topic based on text data, as shown in Formula (12).

$$DT = \alpha \times NI_t^z + \beta \times B + \chi \times R \tag{12}$$

where $DT$ is the value for detecting disruptive technology topics, and $\alpha$, $\beta$, and $\chi$ are tuning coefficients that adjust the weights of each indicator. This formula comprehensively considers the characteristics of disruptive technology topics and normalizes them, adjusting the parameters of $DT$ to obtain a detection value that reflects the disruptive nature of the technology topic. If the detection value is greater than the set parameter, it can be considered that the topic is a disruptive technology topic.

### 3.5. Disruptive Technology Topics Trend Prediction

This article considers the impact of time on the development of a topic and uses time series analysis to analyze the trend. It selects the ARIMA model to make a prediction on the development of the disruptive technology topics. The model views the data sequence formed by the prediction object over time as a random sequence and predicts future values based on the past and present values of the time series. The specific steps are as follows:

(1) In the process of using the LDA model to identify topics, the annual probability distribution of each topic is obtained through a custom function, thereby obtaining the topic change time series data.

(2) Plot the data and observe whether it is a stationary time series. For non-stationary time series, perform a d-order difference operation to convert it into a stationary time series. If it is a stationary series, use the ARMA model directly.

(3) After the second step, a stationary time series has been obtained. To obtain the autocorrelation coefficient ACF and partial autocorrelation coefficient PACF of the stationary time series, the optimal order and number of orders are obtained through analysis of the autocorrelation graph and partial autocorrelation graph.

(4) Construct the ARIMA model with the parameters obtained above and perform a residual test on the model to verify that the obtained model is consistent with the observed data characteristics. If it is not consistent, return to step three to adjust the parameters again.

## 4. Experiment and Analysis

### 4.1. Data Collection and Preprocessing

This article's patent data comes from the China Intellectual Property Network's patent database. Based on literature research and expert knowledge, the search formula for patents in the energy technology field was ultimately determined to be SU = ('energy' + 'energy technology' + 'energy technology') OR SU = ('energy') AND (TKA = ('solid fuel' + 'liquid fuel' + 'gas fuel' + 'hydro energy' + 'nuclear energy' + 'electric energy' + 'solar energy' + 'biomass energy' + 'wind energy' + 'ocean energy' + 'geothermal energy' + 'renewable energy' + 'new energy')), with a filing time of 2012 to 2021. A total of 124,129 patents were obtained through screening, forming the initial patent set. Relevant information such as patent number/application number, title, abstract, International Patent Classification number (IPC), and application date was extracted to complete the data collection.

The data was preprocessed as follows: the abstract text was cleaned and the package in Python was used for word segmentation, filtering punctuation and numbers, and denoising. This included removing stop words and useless high-frequency interference words, and extracting word stems. To improve word segmentation, a dedicated custom dictionary was established to accurately recognize special terms in the energy technology field, such as "smart grid" and "integrated energy service", during the word segmentation process. At the same time, high-frequency words with no actual meaning were added to the stop word list to reduce their impact on topic extraction, such as "the present invention" and "disclosure".

### 4.2. Disruptive Technology Topics Identification

Perplexity is currently the most common evaluation metric in natural language processing, with lower perplexity indicating better model performance [35]. However, perplexity decreases with an increase in the number of topics, and having too many topics can cause the generated model to overfit. Therefore, relying solely on perplexity to judge a model's quality is inadequate. Coherence is introduced as an additional measure to further evaluate the quality of topic models and provide additional guidance when selecting the optimal number of topics. Coherence measures the consistency and interpretability of topics by comparing the co-occurrence of words in each topic. Higher coherence indicates that the words within a topic are more similar and semantically related, making it easier to explain and understand [36]. Considering both perplexity and coherence can help choose the optimal number of topics and improve the quality and performance of topic models. Therefore, the optimal number of topics has a relatively low perplexity and a relatively high coherence. From Figure 2, it can be observed that when the number of topics is 30, coherence reaches its peak and perplexity is relatively low. Therefore, setting the number of topics to 30 is the optimal choice.

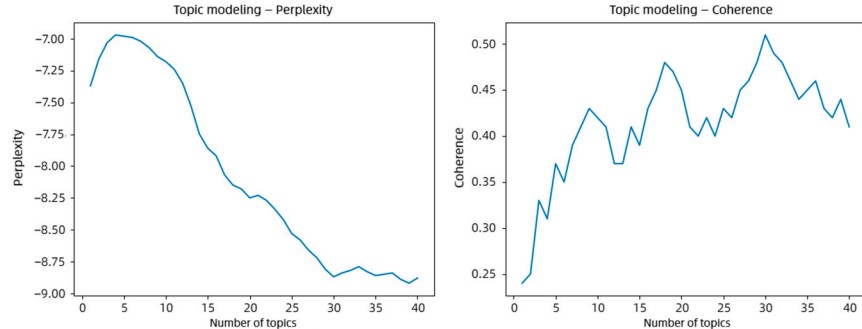

**Figure 2.** Changes in perplexity and coherence as the number of topics varies.

Based on the patent abstracts, a fused feature vector is constructed and technology topics are clustered. The pyLDAvis toolkit is used to visualize the topic clustering results for easier observation and analysis, as shown in Figure 3. The bubbles on the left represent

topics, with the size indicating the frequency of the topic. The distance between the bubbles indicates the similarity between the two topics, and overlap between the bubbles indicates overlap in the characteristic words of the topics. The top 30 characteristic words of each topic are shown on the right, with the light blue indicating the frequency of the word in the entire document and the dark red indicating the weight of the word in the topic. The parameter $\lambda$ at the top right can change the relevance of the words to the topic. When $\lambda$ is close to 0, the words that are specific to the topic are more relevant to the topic; when $\lambda$ is close to 1, the frequently appearing words in the topic are more relevant to the topic. By summarizing the meanings of the words and the relevance of the corresponding topics, the meaning of the topic can be inferred.

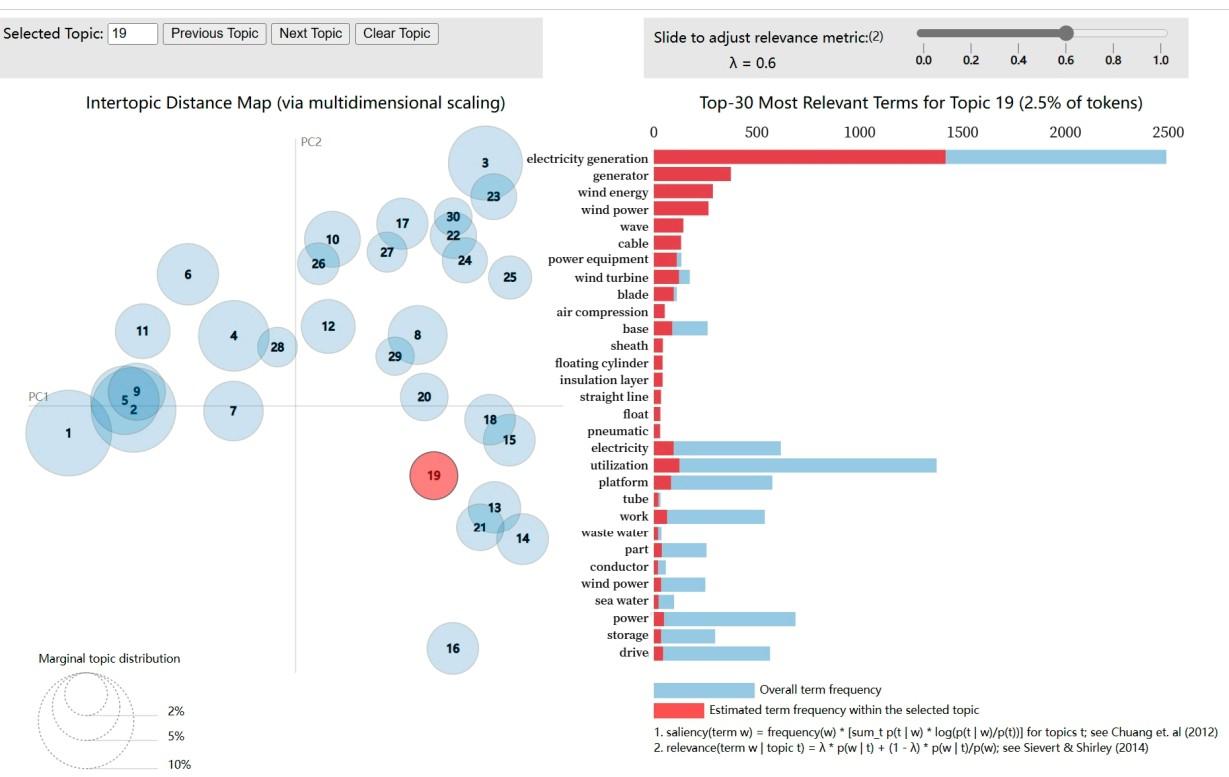

**Figure 3.** Visualization analysis of topic clustering [37,38].

According to (9), (10), and (11), disruptive technology topics usually show a certain degree of novelty, technological breakthrough, and potential scientific impact. Therefore, considering these three indicators and normalizing them, the formula for detecting disruptive technology topics is obtained. It was found in the experiment that, when the parameters $\alpha$, $\beta$, and $\chi$ are set to 0.25, 0.40, and 0.35, respectively, there are fewer noise topics in the detection of disruptive topics and the parameter fitting is good. In this article, a detection value greater than 0.9 is set as a disruptive technology topic, and a total of 10 disruptive technology topics are finally identified. This is shown in Table 1, specifically.

Based on the topic of disruptive technology, select the top 30 topic words for each topic for visual analysis, and draw 10 topic word cloud maps for disruptive technology, as shown in Figure 4.

**Table 1.** Disruptive technology topics identification.

| Serial Number | Topic Characteristic Words | Topic Detection Value | Serial Number | Topic Characteristic Word | Topic Detection Value |
|---|---|---|---|---|---|
| Topic 1 | park, energy, dispatch, system optimization, efficiency, . . . | 0.831 | Topic 16 | hydrogen production, subsystem, electric energy, electrolysis, hydrogen gas, . . . | 0.937 |
| Topic 2 | energy management, internet, information, transmission, terminal, . . . | 0.931 | Topic 17 | maintenance, integrated, temperature sensor, centralized, adaptability, . . . | 0.863 |
| Topic 3 | organization, support, frame, activity, spring, . . . | 0.863 | Topic 18 | hydrogen storage, material, alloy, container, solid state, . . . | 0.916 |
| Topic 4 | energy storage, power, storage battery, capacitor, inductor, . . . | 0.963 | Topic 19 | electricity generation, generator, wind energy, wind power, wave, . . . | 0.949 |
| Topic 5 | optimization, model, coordination, algorithm, energy network, . . . | 0.863 | Topic 20 | hot dry rock, crack, reservoir, fluid, cleft, . . . | 0.927 |
| Topic 6 | power, voltage, test, threshold, current, . . . | 0.842 | Topic 21 | neutron, melting, valve, powder, raw material, . . . | 0.852 |
| Topic 7 | new energy, unit, power grid, electricity, capacity, . . . | 0.875 | Topic 22 | flexible, insulation, vacuum, magnetic field, coil, . . . | 0.876 |
| Topic 8 | fuel cell, nuclear pile, vehicle, dynamic battery, electrode, . . . | 0.921 | Topic 23 | internal combustion engine, cold water, screw, sleeve, slide rail, . . . | 0.891 |
| Topic 9 | evaluation, indicator, evaluation, analysis, planning, . . . | 0.869 | Topic 24 | solar energy, photovoltaic, electrolytic cell, battery, storage battery, . . . | 0.954 |
| Topic 10 | signal, sensor, monitoring, circuit, controller, . . . | 0.874 | Topic 25 | engine, three-way, guide tube, current guide, terminal, . . . | 0.832 |
| Topic 11 | data, distribution network, node, failure, database, . . . | 0.874 | Topic 26 | monitoring, cloud, platform, remote, sensing, . . . | 0.861 |
| Topic 12 | unit, direct current, transformer, bus, alternating current, . . . | 0.841 | Topic 27 | shell, laser, nuclear fusion, target, inertia, . . . | 0.943 |
| Topic 13 | biomass, boiler, box, particle, flue gas, generation, . . . | 0.932 | Topic 28 | prediction, neural network, probability distribution, error, sequence, . . . | 0.864 |
| Topic 14 | air, fuel, pipeline, separator, thermo-electricity, . . . | 0.859 | Topic 29 | region, catalyst, electrochemistry, activity, conductivity, . . . | 0.881 |
| Topic 15 | heat exchanger, heat pipe, heating, circulating pump, thermal efficiency, . . . | 0.838 | Topic 30 | subject, electricity use, module, battery pack, terminal plate, . . . | 0.859 |

The higher the frequency at which a feature word appears in the text data source, the larger the space it occupies in the word cloud. For example, in Topic 4, words such as energy storage and power occupy most of the word cloud, so these words are high-frequency topic feature words for Topic 4. Based on the characteristics of the above word cloud, this paper extracts the top 10 feature words for each disruptive technology topic, as shown in Table 2.

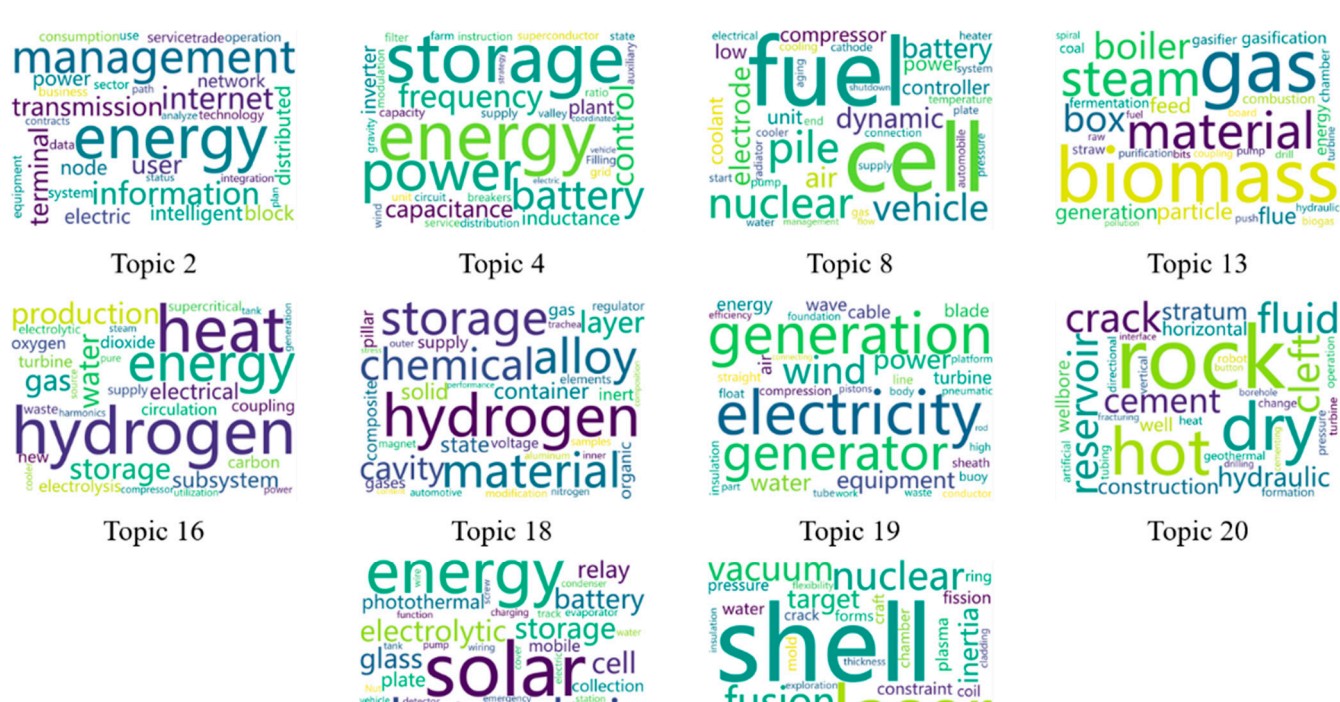

**Figure 4.** Word cloud map for disruptive technology in the field of energy technology.

**Table 2.** Disruptive technology topics in the field of energy technology.

| Serial Number | Topic Characteristic Words (TOP10) |
|---|---|
| Topic 2 | energy management, internet, information, transmission, terminal, user, intelligent, electric power, block, node |
| Topic 4 | energy storage, power, storage battery, capacitance, inductance, power plant, inverter, power supply, capacity, superconductor |
| Topic 8 | fuel cell, nuclear pile, vehicle, dynamic battery, electrode, air compressor, coolant, power, controller, unit |
| Topic 13 | biomass, boiler, box, particle, flue gas, generation, gas, feed, gasification, biomass energy |
| Topic 16 | hydrogen production, subsystem, electrical energy, electrolysis, hydrogen gas, carbon dioxide, oxygen, hydrogen energy, new energy, gas turbine |
| Topic 18 | hydrogen storage, material, alloy, container, solid state, hydrogen supply, hydrogen gas, composite material, organic, pillar |
| Topic 19 | electricity generation, generator, wind energy, wind power, wave, cable, power equipment, wind turbine, blade, air compression |
| Topic 20 | hot dry rock, crack, reservoir, fluid, cleft, cement, hydraulic, stratum, construction, horizontal well |
| Topic 24 | solar energy, photovoltaic, electrolytic cell, battery, storage battery, glass, relay, photothermal, back plate, temperature |
| Topic 27 | shell, laser, nuclear fusion, target, inertia, component, nuclear fission, constraint, plasma, core |

Topic 4's top 10 topic words are "energy storage", "charge and discharge", "battery", "capacitance", and "inductance", among others. From the feature words associated with this topic, it can be determined that the topic focuses on the field of energy storage. Based on the International Patent Classification, this technological topic is "energy storage technology", a technology for storing electrical energy. Energy stored through energy storage technology can be used as an emergency energy source, charged during times of low demand on the grid, and discharged during times of high demand to smooth out fluctuations. The

accuracy of this conclusion can be further verified through text analysis of the search results. For example, patent number CN202110660259.9 describes a hybrid energy storage system that can provide a stable power output to the grid from a transformer station, while also improving the deep discharge effect of a lead-acid battery pack, to a certain extent reducing the number of charge and discharge cycles and effectively extending the battery's service life.

Topic 19's top 10 topic words are "generation", "generator", "wind energy", "wind power", and "wave", among others. From the feature words associated with this topic, it can be determined that the topic focuses on the field of utilizing offshore wind energy. Based on the International Patent Classification, this technological topic is "offshore wind energy technology", a technology that utilizes wind energy for electricity generation by building wind power generators in the water. It has advantages such as not occupying land, high wind speeds, low sand and dust, large power output, stable operation, and zero dust emissions. The accuracy of this conclusion can be further verified through text analysis of the search results. For example, patent number CN202111614982.X describes a combined generator group and offshore wind power system that fully utilizes offshore wind energy and wave energy to increase the utilization rate of offshore energy, increase power generation, and reduce the cost of electricity generation.

Topic 24's top 10 topic words are "solar energy", "photovoltaic", "electrolysis", "battery", and "storage battery", among others. From the feature words associated with this topic, it can be determined that the topic focuses on the field of utilizing solar energy. Based on the International Patent Classification, this technological topic is "photovoltaic solar energy technology", a technology that converts light energy into electrical energy directly using the photovoltaic effect at the semiconductor interface. The advantages of photovoltaic power generation include fewer geographical restrictions, safety, reliability, low noise, low pollution, no need for fuel consumption, and a short construction period. The accuracy of this conclusion can be further verified through text analysis of the search results. For example, patent number CN202011384301.0 describes a photovoltaic solar energy generation and energy storage system with high light energy utilization, which improves the efficiency of photovoltaic cells in converting light into electricity and increases the energy storage efficiency of the power plant.

Similarly, based on the International Patent Classification, the disruptive technology topics corresponding to topics 2, 8, 13, 16, 18, 20, and 27 are "energy internet management technology", "fuel cell technology", "biomass energy utilization technology", "hydrogen production technology", "hydrogen storage technology", "geothermal engineering technology", and "nuclear energy and safety technology", as shown in Table 3.

**Table 3.** Identification of disruptive technology topics in the field of energy technology.

| Serial Number | Disruptive Technology Topics Name |
| --- | --- |
| Topic 2 | energy internet management technology |
| Topic 4 | energy storage technology |
| Topic 8 | fuel cell technology |
| Topic 13 | biomass energy utilization technology |
| Topic 16 | hydrogen production technology |
| Topic 18 | hydrogen storage technology |
| Topic 19 | offshore wind energy technology |
| Topic 20 | geothermal engineering technology |
| Topic 24 | photovoltaic solar energy technology |
| Topic 27 | nuclear energy and safety technology |

To verify the reliability and superiority of the identification method, this study compares the effects of using LDA model, LDA2Vec model, and the improved LDA2Vec model proposed in this study for topic identification. The dataset was trained and tested using

ten-fold cross-validation, and the final F1 values were 74.1%, 83.7%, and 88.5%, respectively. The comparison of the identification results is shown in Table 4.

**Table 4.** Comparison of disruptive technology topics identification results.

| Identification Method | Precision | Recall | F1 Value |
|---|---|---|---|
| LDA mode | 72.3 | 75.9 | 74.1 |
| LDA2Vec model | 82.1 | 85.3 | 83.7 |
| Improved LDA2Vec model | 87.4 | 89.6 | 88.5 |

It can be seen that the improved LDA2Vec model has the best performance for topic identification, with higher precision, recall, and F1 values than the other models. This model combines topic features and semantic features, incorporating global semantic features of text such as global semantics, word order information, and deep semantic association information, to a certain extent overcoming the problem of semantic deficiency in text and more comprehensively and accurately expressing the semantic information of the text vector. In addition, the model introduces the idea of the K-means algorithm into the clustering process and iteratively optimizes the clustering effect of the topics through a greedy strategy, resulting in good performance for identification.

*4.3. Trend Prediction of Disruptive Technology Topics*

In order to forecast the trend of disruptive technology topics, this paper constructs a time series data and uses the ARIMA model to forecast the trend. The reason for using the ARIMA model is that the model is a classic time series forecasting model that only requires endogenous variables and does not require other exogenous variables. In addition, compared to the time series data obtained through the frequency change of keywords, this paper obtains more stable time series data based on the custom function in the topic recognition model used earlier, which makes the topic variable closely related to its previous variables, so the time series change of disruptive technology topics is predictable and suitable for time series forecasting models, which meets the high stability requirements of the ARIMA model for time series data, and the use of the model to forecast trend changes is applicable. In summary, the time series data of disruptive technology topics obtained in this paper is shown in Figure 5.

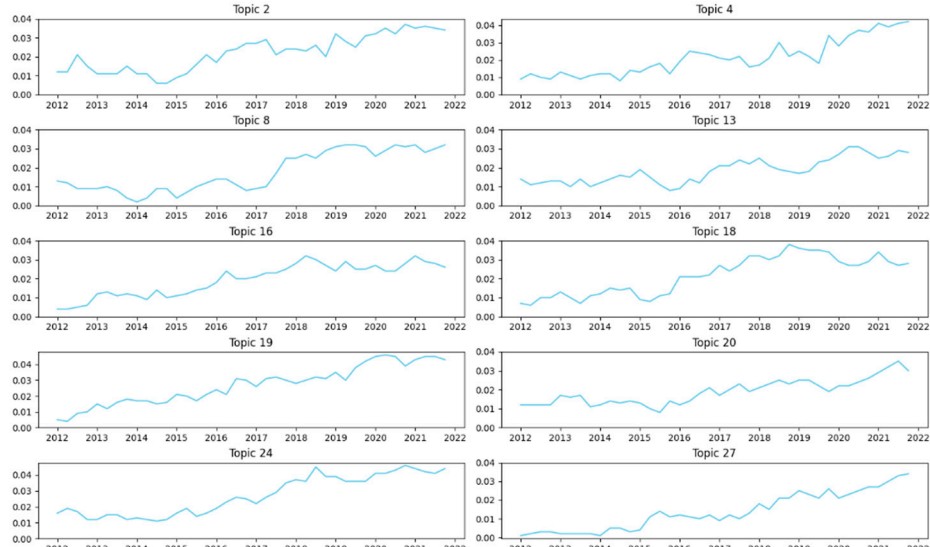

**Figure 5.** Time series of disruptive technology topics in the energy technology field (2012–2021).

Due to space limitations, this paper provides a detailed analysis of the ARIMA model construction process for Topic 4, which has the highest trend degree. The results for the

remaining nine topics are presented without detailed analysis. The time series is second-order differenced which, compared to first-order differencing, only expands the significance level. Therefore, first-order differencing can be used to obtain a stationary time series, with d = 1. The autocorrelation function (ACF) and partial autocorrelation function (PACF) distribution graphs of the topic trend degree sequence are then generated, as shown in Figure 6. Both ACF and PACF exhibit two-order truncation, which can estimate p and q to be 2.

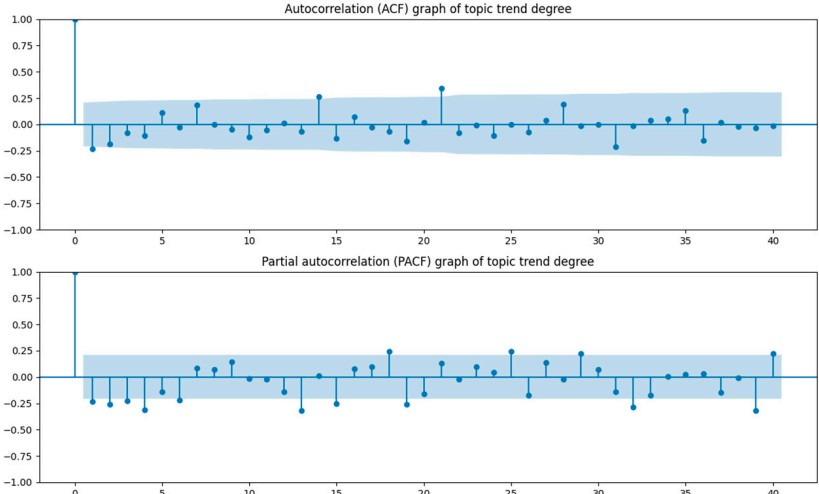

**Figure 6.** Model determination of p and q.

In order to accurately determine the parameters of the ARIMA model, this paper combines the BIC criterion to determine the optimal parameters. The optimal parameters of the ARIMA model for topic 4 are (0, 1, 1). The residual normality test found that the residual distribution histogram is normally distributed and has a good fit for random error. It can be considered that the model extracts all the predictable parts, and the test results are independent, as shown in Figure 7.

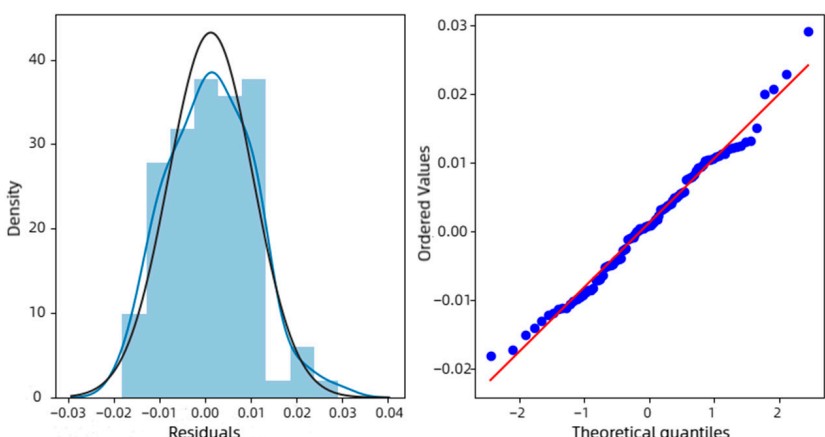

**Figure 7.** Model testing results.

### 4.4. Results and Discussion

Through the model verification of topic 4, it can be seen that the prediction results of the ARIMA model are robust. Therefore, all disruptive technology topics are predicted and analyzed, and the topic trend degree time series prediction graph is drawn, as shown in Figure 8.

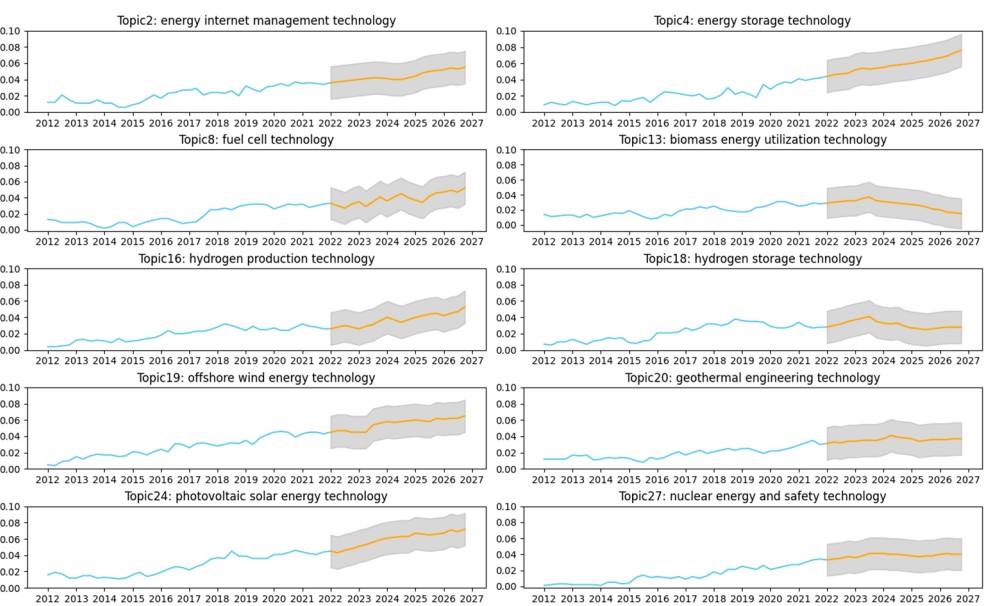

**Figure 8.** Disruptive technology topics trend prediction.

Based on the forecast of the trend of the 10 disruptive technology topics, the specific analysis is as follows:

Disruptive technology topic 2 is energy internet management technology. Under the carbon neutral scenario, China's future energy structure will shift towards a non-fossil fuel-based structure, requiring the construction of a multi-faceted, smart, safe, and flexible energy system to adapt to it. This means that the importance of research in energy internet technology and smart energy system technology will increase, building the next generation smart grid. At the same time, modern energy systems inevitably need the support of interdisciplinary scientific and technological cross-fusion of big data mining, information flow management, decision optimization, etc., and relevant basic research and technological research and development will continue to be of concern. Therefore, based on Figure 8, it can be seen that this technology topic will show a gradually increasing trend in the next five years.

Disruptive technology topic 4 is energy storage technology. In the past 10 years, energy storage technology has received continuous and widespread attention from the scientific research community. Its rapid progress will become a powerful support for the large-scale development of renewable energy power and electric vehicles, and is an indispensable technological means for realizing the low-carbon transformation of the power system. It is a soft link key node for the future energy system's flexibility, inclusiveness and balance. With the widespread application of energy storage technology in energy production, consumption and low-carbon intelligent transformation, there is an increasingly urgent need to improve the safety, energy density and endurance, and reduce the cost of energy storage batteries. Therefore, based on Figure 8, it can be seen that the trend in this technology topic will show a rapid upward development in the next five years.

Disruptive technology topic 8 is fuel cell technology. A fuel cell is a power generation device that directly converts the chemical energy in fuel into electrical energy through electrochemical reactions, which has advantages of high energy conversion efficiency, low pollution emissions, etc., and its application has been extended to many fields, such as transportation, power, micro-power, and military, but low cost and long life are still the bottleneck problems for fuel cell commercialization. In response to the major demand for clean and efficient new energy power sources, key technologies such as low-cost and long-life electrocatalysts and integrated organic fuel reforming will be the key to solving the key problems of cell performance, life, and cost in the future, and to achieve the large-scale promotion and application of fuel cells. Therefore, according to Figure 8, the trend of this

technology topic will show some fluctuations in the next 5 years, but overall it is in an upward trend.

Disruptive technology topic 13 is biomass energy utilization technology. The development of biomass energy has become the core content of energy transformation and an important way to cope with climate change. Biomass power generation technology is one of the most common and effective ways of biomass energy application. The biomass power generation industry in China is in a steady growth stage. Biomass liquid fuels will grow moderately, clean biomass heating, biogas, and biomass natural gas are expected to achieve commercialization and scale, and the biomass energy development and utilization mode will further diversify. Therefore, based on Figure 8, it can be seen that this topic shows a downward trend, and the main reason may be that the development of the technology has matured.

Disruptive technology top 16 is hydrogen production technology. As the scale of renewable energy generation increases rapidly, the grid's ability to absorb the energy becomes a bottleneck. Hydrogen production through electrolysis of water can effectively reduce the abandonment rates of wind and solar energy, and solve the problem of the scale of photovoltaics and wind power being limited by consumer demand and grid absorption capacity. At the same time, hydrogen production from renewable energy sources will also become a sustainable source of green hydrogen in the future. The focus of research is on high-efficiency, clean hydrogen production technology using fossil energy, as well as low-cost hydrogen production technology based on renewable energy sources. The ultimate goal and direction of hydrogen production technology is to achieve carbon-free hydrogen production. Therefore, according to Figure 8, the trend in this technology topic is expected to fluctuate somewhat in the next 5 years, but is overall an upward development trend.

Disruptive technology topic 18 is hydrogen storage technology. Hydrogen energy has the characteristics of high energy density by weight, but low energy density by volume, and the key to the development of its storage and transportation technology is to balance safety and economy while increasing the energy density of hydrogen. Currently, high-pressure gas and liquid hydrogen storage are the main forms of application, but they are not the best options. Organic liquid hydrogen storage has great potential due to its safety, convenience, and high density. In the short term, we should focus on technical breakthrough in compact, lightweight, low-cost, high-pressure hydrogen storage technology to meet the initial commercial application. Therefore, as can be seen in Figure 8, the trend of this technology topic will show a trend of first increasing, then decreasing, and finally leveling off in the next five years.

Disruptive technology topic 19 is offshore wind power technology. China has made a breakthrough in the development of wind power technology, from onshore to offshore, from key components and machine design and manufacturing to wind farm development, and its power generation development scale remains the world leader. With the rise of offshore wind power, the development of large-capacity offshore wind turbines has become an important trend. At the same time, offshore wind farms will develop towards large-scale and deep-sea, and the degree of specialization of operation and maintenance equipment will continue to improve. With the progress of information technology, using big data for self-learning optimization of the turbine can significantly improve power generation efficiency, and intelligent data can predict the failure of turbine components, optimizing the operation and maintenance of offshore wind farms. Therefore, as can be seen in Figure 8, this technology topic in the next five years will show a gradual upward trend.

Disruptive technology topic 20 is geothermal engineering technology. Geothermal energy development, characterized by its sustainability and high-efficiency recycling, has the potential to reduce greenhouse gas emissions and improve the ecological environment, and it is a new direction for energy structure transformation. Currently, the best mining conditions are shallow geothermal anomalous zones, and as technology continues to develop, the cost of geothermal energy development will continue to decrease. Developing key technologies, completing equipment research and development, and conducting efficient

development of dry hot rock geothermal energy, etc.. will be the future development trends for this technology topic. Therefore, as can be seen in Figure 8, the trend in this technology topic will be relatively flat in the next five years, basically stable.

Disruptive technology topic 24 is solar photovoltaic technology. Currently, China's solar photovoltaic industry has begun to take shape, with huge potential for development. From an industrial chain perspective, as the entire photovoltaic industry is still in a rapidly developing stage, progress in production technology and processing technology is very rapid, pushing for continuous updates and replacements of photovoltaic equipment. From the application perspective, increasing photovoltaic conversion efficiency and realizing the operation of solar power generation and existing grid connection are directions that can fully tap the potential of solar photovoltaics. With increasing tension concerning the world's energy supply and the continuous development of photovoltaic technology, the maximization of the development and use of solar energy will be the direction of this energy technology field. Therefore, as can be seen in Figure 8, this technology topic in the next five years will show a gradual upward trend.

Disruptive technology topic 27 is nuclear energy and safety technology. Nuclear energy includes nuclear fission energy and nuclear fusion energy. Through the continuous improvement of nuclear energy technology, the design and construction of standardized and modular reactors, and the development of advanced nuclear energy and nuclear fuel cycle technology, it is possible to achieve optimal utilization of nuclear resources and minimize radioactive waste. The impact of the Fukushima nuclear accident has led to higher demands for nuclear power safety, which has led to the mainstream choice of mature, advanced, and economically safe third and third-generation technology for new nuclear power projects in the short term, while nuclear fusion technology will be the long-term direction of nuclear energy technology development. Therefore, as can be seen in Figure 8, the trend in this technology topic in the next five years will show a relatively stable development.

*4.5. Result Validity Analysis*

This article uses expert interviews and document verification methods to validate the results. According to the *China Engineering Science and Technology 2035 Development Strategy Research Report* of the Chinese Academy of Engineering, clean and efficient development and utilization of fossil energy resources, exploration and development of unconventional oil and gas resources and deep-sea oil and gas resources, independent and innovative nuclear power technology and nuclear waste treatment technology, intelligent power grids and new energy storage technology, and large-scale use of renewable energy technology are the priority technology directions for future development [39]. In addition, the Wang Hongwei team focuses on the engineering and technology needs of future key industries and fields, and invites strategic experts in the field to evaluate the prospects of technology development. It is found that, in order to meet future social demand for safe, green, efficient, and diverse energy, the energy industry should also strengthen research and development in areas such as energy internet, fuel cells, intelligent optimized drilling systems, comprehensive utilization of renewable energy, nuclear energy utilization, intelligent power grids and energy storage technology [40]. The above analysis is basically consistent with the prediction of disruptive technology topics in the energy technology field obtained in this article, which proves the validity of the results.

## 5. Conclusions

The analysis of disruptive technology topics above shows that, with the continuous advancement of technology, the energy sector is undergoing profound changes, with many new technologies and applications emerging. These new technologies will have far-reaching impacts on the future development of the energy industry.

Firstly, from an overall trend perspective, most of the disruptive technology topics are showing an upward trend in the next five years, indicating that these disruptive

technologies will become hot topics in future development. The rapid development of these technologies will help promote China's energy structure transformation from being fossil fuel-based to non-fossil fuel-based.

Secondly, it can be seen that energy internet management technology, energy storage technology, and solar photovoltaic technology show a relatively clear upward trend. The development of energy internet management technology and smart energy system technology can make future energy systems more intelligent, secure, flexible, and diversified. The rapid progress of energy storage technology will be a strong support for the large-scale development of renewable energy electricity and electric vehicles. The continuous improvement of solar photovoltaic technology's photoelectric conversion efficiency and the direction of achieving grid-connected operation with the existing power grid can fully tap the potential for solar photovoltaic development.

In addition, the development trends and key research directions of each technology topic have intersections and commonalities. For example, energy storage technology, hydrogen production technology, hydrogen storage technology, and other topics are all aimed at solving the problems of unstable and uncontrollable renewable energy generation, improving energy conversion efficiency, and achieving energy storage and balance, with strong synergies. The development of energy internet management technology, offshore wind energy technology, geothermal engineering technology, and other topics are all aimed at promoting the development and utilization of clean energy and achieving sustainable energy development, with strong synergies.

At the same time, the development of each topic is also influenced by policy and market environments. For example, due to the promotion of national policies and increasing market demand, the development of solar photovoltaic technology, offshore wind energy technology, and other topics is showing a gradually upward trend. The development of topics such as nuclear energy and safety technology and biomass energy utilization technology are constrained by policy and market factors and may face certain development difficulties.

In summary, the identification and prediction of disruptive technology topics provide important references for enterprises, research institutions, and governments, and are of great significance for promoting technological innovation and development and driving sustainable economic and social development. All parties should closely monitor the development dynamics and trends of disruptive technologies, strengthen cooperation, jointly promote research and application of disruptive technologies, and achieve sustainable development of the economy, environment, and society.

### 5.1. Possible Research Contributions

Firstly, this paper employs an improved LDA2Vec model that integrates topic and semantic features, including global semantic, lexical ordering information, and deep semantic correlation information. This overcomes the problem of missing semantic information in the text and can express the semantic information of text vectors more comprehensively and accurately. In comparison experiments with general models, the precision, recall, and F1 values of this model are all higher, indicating a more accurate identification of technological topics.

Secondly, in the process of identifying topics, this model can obtain the annual probability distribution of each topic, thereby obtaining stable topic trend time series. Based on this, a more objective prediction model of the trend of disruptive technology topics is constructed, improving the scientificity and accuracy of ARIMA model predictions.

Finally, in order to verify the correctness of the conclusions, this paper employs expert interviews and data verification methods to compare the results, and the two are consistent, thereby validating the effectiveness and feasibility of the proposed method. Therefore, this paper effectively grasps the development trend of disruptive technology topics in the energy technology field and has important guiding significance for the future development layout of relevant industries.

*5.2. Research Shortcomings and Future Research Directions*

Firstly, it is important to note that the development trend of technology is only a predictive result, and the actual development may be influenced by various factors, including policies, markets, and technological breakthroughs. Therefore, the combination model proposed in this paper still has limitations, such as the topic recognition model being affected by the number of training times, model parameters, and data set size, which can be further optimized in the future.

Secondly, in terms of the predictive model, this study only used the ARIMA model for prediction. Future research directions could consider using more advanced predictive models, such as machine learning, deep learning, and ensemble algorithms for comparative analysis. Machine learning models can automatically learn the relationships between disruptive technology topics while identifying them, improving the accuracy and stability of the predictive model. Deep learning models can better understand the semantic information of the text, improving the model's prediction ability. Ensemble algorithms can combine the predictive results of multiple models, reducing the prediction error of the model, and improving the prediction accuracy and stability.

Finally, future research directions can also explore the evolution mechanism of disruptive technology topics, understanding the relationships between technology topics and their evolutionary paths. By analyzing the relationships between topics, the evolution of technology can be more accurately predicted, which will provide more valuable insights into the development trends of disruptive technologies.

**Author Contributions:** Conceptualization, methodology, software, validation, formal analysis, and writing—original draft preparation, M.X. and K.L.; resources, investigation, writing—review and editing, and supervision, D.F. All authors have read and agreed to the published version of the manuscript.

**Funding:** This research was supported by the National Social Science Foundation of China (22CTQ028).

**Institutional Review Board Statement:** Not applicable.

**Informed Consent Statement:** Not applicable.

**Data Availability Statement:** The authors would like to thank the editors and anonymous reviewers for their thoughtful and constructive comments.

**Conflicts of Interest:** The authors declare no conflict of interest.

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
