# Peer review of "Identifying and Predicting Trends of Disruptive Technologies: An Empirical Study Based on Text Mining and Time Series Forecasting"

_sustainability, doi:10.3390/su15065412_

Round 1

Reviewer 1 Report

This study used various algorithms to develop a text analysis model for patent documents and predict future trends, and confirmed its performance through expert verification. The technology can be applied to various fields, so it is considered a suitable paper for sustainability. However, there is a question about the selection of the optimal number of topics, which is the initial basis of the analysis process of the paper, and it is judged that publication is possible through supplementation of this. It is recommended as a major revision to check the supplementation of the comment.

1. Perplexity was used to calculate the appropriate number of topics. What is the difference between degree of confusion and consistency size, which are basically calculated for Bag of Words (BoW)? (Figure 2)

2. Basically, when calculating the number of topics in BoW through data preprocessing, the degree of confusion continues to change depending on the data used. Is the optimal number of topics (30) determined simply through a single process?

2.1 Do you keep getting the same result when counting the number of topics?

2.2 If the same result is not continuously shown, the reliability of the subsequent analysis is lowered, so an accurate answer is required.

3. The main words in Topic 19 are in the order of power generation, generator, wind energy, and wind power. The results shown in figures 3 and 4 and tables 1 and 2 seem to be partially different.

3.1 Electricity generation occupies a really high proportion in figure 2, why is it not highlighted in wordcloud, etc.?

Reviewer 2 Report

1、For each dictionary, why only one topic with the highest probability is selected instead of multiple topics?

2. Would it be more obvious to compare the two figures in Figure2 on a single figure?

3. What is the connection between the content of the abstract and the text? What is the inverted U-shaped effect, the positive spatial transmission effect?

4. Clearly written contributions is needed.

Reviewer 3 Report

The paper under consideration covers an important topic in the literature.

It is well motivated in the introduction and the methods used in the empirical analysis are adequate and allow to obtain interesting results.

My overall opinion about this study is positive. I have however some concerns that in my opinion must be addressed before publication, namely:

- I feel the lack of a literature review - it could be as a separate section or further development in the introduction.

- regarding the evidence produced, I would like to see some more in-depth discussion of some results, namely through a closer link with previous studies on related topics.

Reviewer 4 Report

1.       Authors have addressed limitations of standalone LDA model, such as ignoring semantic relationships between words, affecting the accuracy of recognition and hindering subsequent trend prediction. Therefore, they have proposed LDA2Vec model that combines global semantic word vectors and text-based topic vectors to overcome the issues. They have also introduced K-means algorithm for performance improvement. Then they have used time series model for trend prediction.

2.       What are the other methods of studying disruptive technology?

3.       More literature reviews are required for text mining and time series prediction models.

     They should refer machine learning, deep learning, and ensemble algorithms for prediction.

4.       The manuscript is well organized though certain improvements are needed.

5.       Authors should have used full form of abbreviations when using for 1st time (line 66).

6.       Authors have described disruptive technologies in a single sentence only (line 52, 53, 54) though this needs more clarification as this is the main area of their research. They have used the term ‘disruptive technologies’ many a time, instead they may use any abbreviation.

7.       Equation numbers may be cited inside the text.

Round 2

Reviewer 1 Report

The logic and consistency of the paper have been secured, and the comments seem to have been appropriately reflected.

Reviewer 2 Report

This paper proposed a topic recignition and trend prediction method and provide more valuable insights into the development trends of disruptive technologies.